# The Effect of Exercise Training on Brain Structure and Function in Older Adults: A Systematic Review Based on Evidence from Randomized Control Trials

**DOI:** 10.3390/jcm9040914

**Published:** 2020-03-27

**Authors:** Feng-Tzu Chen, Rachel J. Hopman, Chung-Ju Huang, Chien-Heng Chu, Charles H. Hillman, Tsung-Min Hung, Yu-Kai Chang

**Affiliations:** 1Graduate Institute of Sport, Leisure and Hospitality Management, National Taiwan Normal University, Taipei 10610, Taiwan; alexnewtaipei@gmail.com; 2Department of Psychology, Northeastern University, Boston, MA 02115, USA; r.hopman@northeastern.edu (R.J.H.); c.hillman@northeastern.edu (C.H.H.); 3Graduate Institute of Sport Pedagogy, University of Taipei, Taipei 11153, Taiwan; crhwang@utaipei.edu.tw; 4Department of Physical Education, National Taiwan Normal University, Taipei 10610, Taiwan; cchu042@yahoo.com; 5Department of Physical Therapy, Movement, and Rehabilitation Sciences, Northeastern University, Boston, MA 02115, USA; 6Institute for Research Excellence in Learning Science, National Taiwan Normal University, Taipei 10610, Taiwan

**Keywords:** aerobic exercise, cognitions, magnetic resonance imaging, aging, grey matter, white matter

## Abstract

Accumulating evidence suggests that exercise training is associated with improvements in brain health in older adults, yet the extant literature is insufficient in detailing why exercise training facilitates brain structure and function. Specifically, few studies have employed the FITT-VP principle (i.e., Frequency, Intensity, Time, Type, Volume, and Progression) to characterize the exercise exposure, thus research is yet to specify which characteristics of exercise training benefit brain outcomes. To determine whether exercise training is consequential to cognitive and brain outcomes, we conducted a systematic review investigating the effects of exercise training on brain structure and function in older adults. PubMed and Scopus were searched from inception to February 2020, and study quality was assessed using the Cochrane risk-of-bias tool. A total of 24 randomized controlled trials were included. This systematic review indicates that older adults involved in exercise training may derive general benefits to brain health, as reflected by intervention-induced changes in brain structure and function. However, such benefits are dependent upon the dose of the exercise intervention. Importantly, current evidence remains limited for applied exercise prescriptions (e.g., volume, progression) and future research is needed to clarify the effects of exercise training on cognitive and brain outcomes in older adults.

## 1. Introduction

The worldwide population of older adults is drastically increasing over time [1]. For example, approximately 810 million (11.7%) people were 60 years or older in 2012. By 2050, the projected number of older adults will rise to more than two billion, reaching 21% of worldwide population [2]. The rapid growth of the aging population has brought attention to prevalent age-related cognitive impairments, which are commonly associated with decreased quality of life and increased financial costs of healthcare [3]. Previous research has shown that cognitive performance (e.g., information processing speed, memory, executive function) declines with advancing age [4,5,6]. Age-related impairments also reflect brain atrophy and decreased brain function in several regions, including the prefrontal and temporal cortices, and the hippocampus [7,8,9]. As such, the healthcare field must identify novel therapeutic strategies to combat the rising rate of age-related cognitive decline.

Regular physical activity has emerged as a low-cost, non-pharmacological treatment for slowing the progression of age-related cognitive decline [10,11,12]. Several epidemiological studies have demonstrated a relationship between higher levels of physical activity and reduced risk of cognitive impairment [13,14]. Likewise, exercise training has been observed to improve cognitive functions in community-dwelling healthy older adults [15,16,17,18], and several systematic reviews and meta-analyses of randomized control trials (RCTs) have evidenced these behavioral effects in older populations [19,20,21,22,23,24,25,26].

Although exercise training is associated with improvements in cognitive function, research is yet to fully understand the parameters under which exercise influences brain structure and function in older adults. The Physical Activity Guidelines for Americans [27] updated the scientific knowledge regarding the benefits of exercise across the lifespan and further emphasized specific characteristics of exercise training that lead to changes in brain health. Further, the American College of Sports Medicine [28] provided the FITT-VP principle, which included Frequency (number of exercise occurrences), Intensity (difficulty of the exercise bout), Time (length of intervention), Type (mode of the exercise bout), Volume (total amount of exercise), and Progression (change in difficulty in the exercise program over time). Although several current reviews provide evidence of a relationship between physical fitness and white matter volume [29] or aerobic exercise and hippocampal volume [30], research has yet to determine if a relationship between specific exercise characteristics and changes in brain structure and function exist. In addition, recent reviews have only examined the relationship between cognitive changes and exercise time (i.e., length) [31], but have not examined if other FITT-VP characteristics of exercise training relate to changes in brain structural and functional outcomes.

Accordingly, previous evidence remains insufficient to summarize whether the aspects of exercise training, characterized via the FITT-VP principle, relate to changes in brain structure and function. To determine whether exercise training is consequential to brain outcomes, we conducted a systematic review investigating the relationship between exercise training and brain structure and function using magnetic resonance imaging (MRI) and functional MRI (fMRI). This review will further expound the neurophysiological mechanisms underlying the effects of exercise training via RCTs in older adults. In achieving these aims, we assessed (1) methodologic quality using criteria for RCTs (i.e., Cochrane risk-of-bias tool), (2) the relationship between exercise training measures and brain structure, and (3) the relationship between exercise training measures and brain function.

## 2. Method

### 2.1. Search Strategy 

A systematic search for this review was conducted in February 2020 by using two databases (PubMed and Scopus) following the Preferred Reporting Items for Systematic Review and Meta-Analysis (PRISMA) statement guidelines [32]. The keywords for searching articles used intervention terms (“exercise” OR “physical activity” OR “physical exercise” OR “physical therapy” OR “physical training” OR “exercise training”), brain terms (“magnetic resonance imaging” OR “MRI” OR “grey matter” OR “white matter” OR “diffusion tensor imaging” OR “voxel-based morphometry” OR “tensor-based morphometry” OR “functional magnetic resonance imaging” OR “fMRI”), and aging terms (“aging” OR “older”). Studies were limited to a human clinical trial design. Additional articles were identified by manually searching and through expert knowledge of relevant papers.

### 2.2. Inclusion/Exclusion Criteria

All articles included in this systematic review met the following criteria: (1) the experimental RCT included baseline and post-intervention brain structure or function outcomes, (2) the experimental group engaged in an chronic exercise intervention, (3) any type of exercise intervention that included planned and structured physical activity designed to increase or maintain physical fitness, (4) participants were 60 years of age or older, (5) or, if other age groups were investigated, our discussion was limited to data only from this age group, (6) participants were cognitively intact, (7) the control group was either an active control such as stretching and balance, light strength-endurance, health education and promotion, or a wait-list control, and (8) articles were written in English.

The exclusion criteria comprised articles that: (1) were a commentary, case report, meta-analysis, or review, (2) were a book section or chapter, or did not include a full-text article (e.g., conference abstracts), (3) did not provide any descriptions of exercise training or training combined other treatments (e.g., cognitive training, supplementary); (4) did not have any brain outcomes or the brain outcomes overlapped across the different studies; or (5) did not have a control group.

The process of study selection was independently reviewed by two authors. They identified eligible studies by inspecting the title, abstract, and keyword. Subsequently, two reviewers inspected the relevant references for eligibility and discussed disagreements in article selection. To ensure that collected data were accurate, each reviewer initially coded 10 studies independently and then compared their data to ensure that any potential differences were minimized. As such, the average inter-rater reliability in coding achieved 95% for all pairs of reviewers. If any remaining disagreements occurred, a third reviewer joined the discussion to reach consensus.

### 2.3. Risk of Bias Assessment 

The quality of included studies was tested using the Cochrane risk-of-bias tool [33], which is commonly used to assess bias in the issue of exercise training and cognitive function in RCTs [34]. We selected six appraisal criteria, including sequence generation, allocation concealment, blinding of participants, blinding of assessors, incomplete outcome data, and selective outcome reporting to evaluate potential source bias. Two reviewers assessed the risk of bias to rate each of the included studies and three ratings (e.g., high risk, low risk, and unclear) were used to judge each study, following Section 8.5 of the Cochrane handbook [33].

### 2.4. Data Collection and Extraction Process

We extracted data based on study characteristics (e.g., sample size, age, gender), grouping, exercise prescription (frequency, intensity, time, type, volume, and progression), as well as the intervention outcomes using MRI techniques (e.g., volume of whole- and specific-brain regions, white matter integrity) or fMRI techniques (i.e., resting-state activation (assessment in the absence of formal task instruction), task-evoked activation (participants were instructed to perform a given task)). In this review, time represents length and session time of the exercise training intervention. The data of exercise prescription components were only extracted when including studies specifically examining the component.

Details of each exercise prescription, frequency (low: 1–2 times per week, moderate: 3–4 times per week, high: 5–7 times per week), intensity (moderate: 40–60% heart rate reserve (HRR)/maximal oxygen uptake (VO_2 max_), 55–70% heart rate max (HR_max_), 50–70% 1 repetition maximum (RM); vigorous: 61–85% HRR/VO_2max_, 71–90% HR_max_, 71–84% 1-RM) [35], length ( short-term: ≦12 weeks, mid-term: 13–24 weeks, long-term: 25–48 weeks, very long-term: >48 week), session time (short time: ≦45 min, moderate time: 46–60 min, long time: >60 min), type (aerobic exercise, resistance exercise, Tai Chi, coordination exercise, dance, combined exercise), volume (low: <150 min of moderate intensity or <75-min of vigorous intensity per week; moderate: 150–300 min of moderate intensity per week; high: ≥300 min of moderate intensity per week) and progression (change in exercise intensity and session time throughout intervention) were coded and discussed.

## 3. Results

### 3.1. Study Selection 

The criteria for study selection (i.e., PRISMA) through the systematic review process is presented in Figure 1. A total of 5863 studies were initially screened using the above search terms through the database search (e.g., PubMed, Scopus) with six additional articles added via other sources. After removing duplicate articles, 2281 articles remained for further screening. Authors subsequently checked titles and abstracts, excluding 119 reviews, 42 book chapters and conference abstracts, and 1970 articles that were not related to the scope of this systematic review. A total of 150 articles were retrieved for full-text screening. After screening, 126 articles were removed because the intervention did not include RCTs (*n* = 13), participants were under 60 years of age (*n* = 4), a brain outcome was not reported (*n* = 4), the research did not include an exercise training intervention (*n* = 74), the study did not meet criteria for an experimental group (*n* = 6) or control group (*n* = 5), or the sample was characterized by neurocognitive diseases (*n* = 19). One study was removed because the brain outcomes overlapped. In total, 24 studies met the criteria and were used in the analyses, including 11 studies on brain structure, 11 studies on brain function, and two studies that assessed both outcomes.

### 3.2. Risk of Bias Analysis of Included Studies

To evaluate the risk of bias, Figure 2 depicts the percentage of bias (i.e., high risk, low risk, unclear) in the studies included in the systematic review. For selection bias, 100% of the studies discussing brain structure or function were deemed low risk in sequence generation randomization into the experimental group (exercise intervention) or control group because all included studies were RCTs. For studies measuring brain structure, 17% of low risk studies and 83% of unclear studies were identified in allocation concealment. For studies measuring brain function, 23% of low risk studies and 77% of unclear studies were identified in allocation concealment. For performance bias, both high and low risk studies were 17% and the unclear studies were 66% for brain structure and 100% of unclear studies for brain function in blinding of the participants. For detection bias, 58% of the low risk and 42% of the unclear studies for brain structure and 38% of the low risk and 52% of the unclear studies for brain function were judged in the blinding of the assessors. For attribution bias, more than 50% of low risk studies for both brain structure and function were identified as having incomplete outcome data. Furthermore, 100% of studies were low risk in selective reporting of the data.

### 3.3. Exercise Prescription Analysis of Included Studies for Brain Structure 

In total, the present review identified 13 studies that examined the effect of exercise intervention on brain structure in older adults, and a summary of these studies appears in Table 1.

#### 3.3.1. Frequency

Of a total of 13 studies, seven studies (54%) reported that older adults were assigned to a moderate frequency (three to four times weekly) intervention [36,37,38,39,40,41,42], three studies (23%) assigned older adults to a low frequency (one to two times weekly) intervention [43,44,45], and one study (8%) assigned older adults to a high frequency (five to seven times weekly) intervention [46]. Two studies (15%) did not report the frequency of exercise training [47,48].

#### 3.3.2. Intensity

The included studies reported several different measurements of intensity during exercise training, including HRR, HR_max_, VO_2 max_, ventilatory anaerobic threshold of oxygen uptake (VO_2VAT_), anaerobic threshold (AT), and RM. Of the included studies in the review, eight studies (62%) used vigorous intensity for exercise training [36,37,38,39,40,43,44,47], and two studies (15%) used moderate intensity [45,48]. Three studies (23%) did not report the intensity of exercise training [41,42,46].

#### 3.3.3. Length

Six studies (46%) indicated that the exercise training intervention occurred for a long-term length of time (25–48 weeks) [37,40,41,44,47,48], three studies (23%) indicated a short-term length (≦ 12 weeks) [38,39,46], three studies (23%) indicated a mid-term length (13–24 weeks) [36,42,45], and one study (8%) indicated a very long-term length (>48 weeks) [43].

#### 3.3.4. Session Time

Six studies (46%) used moderate-time (45- to 60-min) exercise sessions [36,41,42,43,44,46]. In addition, five studies (38%) used short-time (≦ 45 min) exercise sessions [37,38,39,45,47]. One study (8%) did not report the duration of exercise training per session [48] and one study (8%) provided unspecific timing of exercise training (i.e., 30–60 min) [40]. In addition, none of the included studies used a long session time (>60 min) of exercise training.

#### 3.3.5. Type

Seven studies (53%) reported the intervention consisted of aerobic exercise [36,37,38,39,40,45,47], and one study (8%) included aerobic and coordination exercise compared to no exercise training [41]. In addition, two studies (15%) used resistance or strength training [43,44], one study (8%) used Tai Chi [46], one study (8%) used combined exercise [48], and one study (8%) used dance intervention [42].

#### 3.3.6. Volume 

None of the included studies focused on the volume of exercise training.

#### 3.3.7. Progression

Four of the included studies (31%) reported progression in intensity and session time [36,37,40,47].

#### 3.3.8. Brain Regions and Main Interventional Findings

Seven studies (54%) analyzed the effects of exercise training on whole brain volume [36,37,39,43,44,45,46]. In addition, three studies (23%) investigated the effects of exercise training on a specific brain region, such as the hippocampus [38,41,48], and three studies (23%) focused on two or more brain regions [40,42,47].

Altogether, eight studies (62%) suggested that exercise training increased brain structure or reduced grey and white matter atrophy [36,41,42,43,44,46,47,48]. However, five other studies (38%) did not observe any change in brain structure following exercise intervention or between groups [37,38,39,40,45].

### 3.4. Exercise Prescription Analysis for Brain Functioning 

A total of 13 included studies examined the effects of exercise training on brain function, and the summaries of resting-state and task-evoked fMRI studies are described in Table 2 and Table 3, respectively.

#### 3.4.1. Frequency

Nine of the 13 studies (69%) reported that older adults were assigned to a moderate frequency (three to four times weekly) intervention [38,49,50,51,52,53,54,55,56], three studies (23%) reported older adults were assigned to a high frequency (five to seven times weekly) intervention [57,58,59], and one study (8%) reported older adults were assigned to a low frequency (one to two times weekly) intervention [45].

#### 3.4.2. Intensity

Of the included studies in the review, five studies (46%) used vigorous intensity exercise training [38,50,54,55,56], three studies (15%) used moderate intensity [45,49,53], and one study (8%) used moderate-to-vigorous intensity [52]. Four studies (31%) did not report the intensity of exercise training [51,57,58,59].

#### 3.4.3. Length

Seven studies (50%) indicated that the exercise intervention occurred for a short-term length of time (≦12 weeks) [38,49,52,56,57,58,59], three studies (25%) indicated a mid-term length of time (13–24 weeks) [45,50,51], three studies (25%) indicated a long-term length of time (25–48 weeks) [53,54,55], and none of the included studies employed a very long-term length of time (> 48 weeks).

#### 3.4.4. Session Time

Six studies (46%) reported exercise training for a short-time exercise (≦ 45 min) per session [38,45,50,54,55,56], six studies (46%) used a moderate session time (45–60-min) [49,52,53,57,58,59], and one study (8%) did not report the specific session time of exercise training [51]. None of the included studies used a long session time (> 60 min).

#### 3.4.5. Type

Eight studies (62%) reported an intervention using aerobic exercise training [38,45,49,50,51,52,54,55], four studies (31%) used Tai Chi [56,57,58,59], and one study (7%) used coordination exercise compared to aerobic exercise and no exercise training (Voelcker-Rehage et al., 2011). None of the included studies used resistance exercise and combined exercise for examining brain function.

#### 3.4.6. Volume

None of the included studies focused on the volume of exercise training.

#### 3.4.7. Progression

Six of the 13 studies (46%) reported progression in intensity and session time [45,50,52,53,54,55].

#### 3.4.8. Approach and main interventional findings

Nine studies (69%) examined the effects of exercise training on brain function via resting-state fMRI [38,45,49,51,54,55,57,58,59] and four studies (31%) used task-evoked fMRI [50,52,53,56].

Seven studies (54%) showed that exercise training exhibited brain function changes during a resting state, particularly in prefrontal and temporal regions [38,45,49,54,57,58,59] and two studies (15%) showed no change in activation or did not report interventional effects after exercise intervention [51,55]. Four studies (31%) showed changes in brain activation during task-evoked fMRI collection [50,52,53,56].

## 4. Discussion

### 4.1. Summary of Search Results 

The purpose of this systematic review was to summarize the literature that examined the effects of randomized controlled exercise interventions on brain structure and function and to further explore the underlying relationships based on the various exercise prescriptions using the FITT-VP principal in older adults. Overall, a total of 24 empirical articles were identified that explored whether exercise training showed changes in brain structure and brain function (13 datasets for each; two studies employed both structural and functional outcomes). The results of methodological heterogeneity of the included studies indicate that more than 50% were low risk across the four criteria (i.e., sequence generation, blinding of assessors, incomplete outcome data, and selective outcome). We also found that three criteria (i.e., allocation concealment, blinding of participants and assessors) were high risk and unclear for above 50% of the studies, which may have influenced the intervention effects in the observed findings. To ensure whether exercise prescriptions modulated brain outcomes, detailed information regarding the FITT-VP principle was extracted to further describe the relationship between exercise training and brain structure and function. Notably, this is the first systematic review to use FITT-VP as a guiding principal to characterize exercise effects on brain outcomes.

### 4.2. Exercise Training and Brain Structure

The analysis of brain structure revealed that the most common exercise prescription was three to four times per week (frequency), vigorous (intensity), 25–48 week (length), 45- to 60-min (session time), aerobic exercise (type). To understand the effects on brain structure, we classified the included studies according to whether they assessed whole brain structure (global effect) or specific regional structures (selective effect). The results show that seven of the 13 included studies examined the whole brain to assess exercise-induced structural outcomes. Of these studies, four studies showed the exercise intervention group increased brain structure [36,46] or reduced brain volume atrophy [43,44], and three studies found no evidence of a beneficial effect on whole brain volume changes [37,39,45] after intervention. In addition, six of the 13 studies examined specific regional outcomes to understand structural effects associated with exercise training. Three studies showed increased hippocampal volume [41,47,48] and white matter integrity [42] and two studies failed to find differences in cortical thickness [40] or structural volume [38] after intervention.

Approximately half of the included studies assessing brain structure implemented moderate frequency (three to four times per week) exercise training; however, conflicting findings across studies were noted. For example, three studies indicated that older adults engaging in exercise training three times per week increased grey and white matter volume [36], hippocampal volume [41], and white matter integrity [42], yet three studies showed no changes [37,38,40]. In addition, there is evidence suggesting similar structural effects related to low exercise frequency (one to two times per week). For instance, Liu-Ambrose, Nagamatsu et al. [44] suggested both once-weekly and twice-weekly resistance exercise can lead to reductions in whole brain atrophy when compared with a balance and toning group. For high frequency, only one study implemented exercise training five times per week and indicated a positive influence on brain volume [46]. These findings suggest that exercise frequency may not moderate exercise training effects on brain structure outcomes.

In this review, aerobic exercise training was the most common intervention mode, with several studies demonstrating increased brain structure outcomes, including both cortical and subcortical structures, following the completion of the intervention [36,41,47]. Beyond aerobic exercise, other intervention modes (i.e., resistance exercise, Tai Chi) have demonstrated reduced age-related brain atrophy [43,44,46]. When considering intervention intensity, the majority of studies found that vigorous-intensity aerobic and/or resistance exercise training led to increased brain volume [36,41,43,44,47]. These finding are consistent with suggestions from a previous review [31], showing a target zone was sufficient to change brain volume. The present review shows that there was not a target intensity reported for certain exercise types (e.g., Tai Chi, coordination exercise, dance). These types of intervention place demand on motor fitness—including agility and balance—rather than cardiovascular or muscle fitness, which may also benefit brain structure [41,46] and white matter integrity [42]. Taken together, the diversity of exercise types and intensities makes comparisons difficult, thus we recommend using simple measurements to characterize the exercise training program (e.g., rated perceived exertion [RPE], heart rate monitoring, accelerometry) to facilitate comparisons between exercise types.

To quantify the duration of exercise training, three studies evaluated the effects of high length exercise (> 48 weeks) using moderate session time (45–60 min). These studies showed exercise training significantly increased brain volume [41,44] or reduced white matter volume atrophy [43] after intervention. Consistent with the previous literature, these findings suggest that regular exercise training can relate to changes in brain volume [31]. However, one study did not find changes in white matter integrity, including measurements of fractional anisotropy (coherence of the orientation of water diffusion), axial diffusivity (the eigenvalue of the primary axis), or radial diffusivity (the average of the two perpendicular eigenvalues) after exercising 40 min per session for 48 weeks [37]. These findings suggest that exercise length and session time both may influence brain structure in older adults.

Although the present review provides an overview of the relationship between exercise training and brain structure, only four of the 13 studies assessed progression of exercise training; thus, the present review is unable to discuss the effects of this aspect of the FITT-VP principal on brain structure in an older population. Collectively, we recommend that future studies report the dose–response relationships (e.g., exercise volume) so that exercise guidelines for brain structure and function can be better developed.

### 4.3. Exercise Training and Brain Function

Thirteen studies were included in the review that examined brain function outcomes. In our analysis, the most commonly used exercise training prescription was three to four times weekly (frequency), vigorous (intensity), ≦12 weeks (length), and ≦60 min (session time), aerobic training (type). The included studies focused on either resting-state fMRI [38,45,49,51,54,55,57,58,59] or task-evoked fMRI [50,52,53,56]. Taken together, the vast majority of studies (85%) demonstrated changes in brain function following exercise training.

The present review suggests that the frequency of exercise training did not moderate the effects on brain function. For instance, Voss, Prakash et al. [54] showed facilitation of resting-state neural activity and suggested that a 3-day weekly program of aerobic exercise enhanced functional connectivity (i.e., temporal coherence between spatially remote brain regions) in frontal and temporal cortices that underlie higher-level cognitive networks relative to those in a stretching and toning control group. Similarly, an early study showed changes in task-evoked brain activation, as measured via blood oxygen level dependent changes in specific brain regions [50]. In this study, older adults who participated in an exercise training program with moderate frequency (three times per week) exhibited increased activation in the middle frontal gyrus (MFG), superior frontal gyrus (SFG), and superior parietal lobules (SPL), and decreased activation in the anterior cingulate cortex (ACC) during a modified flanker task, which modulated inhibition and selective attention demands. Although only four studies focused on low (one to two times weekly) or high (five times weekly) frequency interventions, the findings from these studies indicate that older adults exhibited resting-state brain activation changes following intervention relative to those assigned to a control group [45,57,58,59]. 

Several studies suggested exercise training at a vigorous intensity facilitates both resting-state and task-evoked brain activation. Specifically, implementing a heart rate reserve (HRR) of 61–85% or a heart rate maximum (HR_max_) of 71–90% induced changes in brain activation [50,54,56]. In addition, the authors assessed a range from 50% to 75% HRR (coded as moderate to vigorous intensity in our review) for a 12-week aerobic exercise intervention and exhibited lower task-evoked activity of the right IFG compared to a control group (i.e., stretch and balance exercises) during the performance of a verbal fluency task, suggesting more efficient use of neural resources during cognitive processing [52]. Four studies did not report exercise intensities during Tai Chi or combined exercise training [51,57,58,59]; therefore, it is difficult to characterize this aspect of the intervention on brain function outcomes. We recommend that future research reports the intensity of exercise training prescriptions during intervention studies to further explore the potential effects of exercise intensity on brain function. 

The overall findings from the included review suggest that aerobic training is a viable strategy to alter brain function [45,50,52,54] and hippocampal perfusion (i.e., blood movement) [38]. A majority of studies reported that increased aerobic exercise training can lead to plasticity of functional outcomes in the aging brain. Importantly, exercise types may differentially affect brain function. For example, Voelcker-Rehage, Godde et al. [53] explored brain function among older adults engaged in a 12-month aerobic and coordination training program and found aerobic exercise training decreased task-evoked activation in the MFG, the left ACC, the left parahippocampal gyrus, and the right superior and middle temporal gyrus, while coordination exercise training increased activation in the IFG, thalamus, caudate, and the superior parietal lobule. In addition, the findings from three other studies are focused on other exercise modes such as Tai Chi and show beneficial effects on functional connectivity [58,59] and fractional amplitude of low frequency fluctuations (regional spontaneous neuronal activity) [57].

The length of exercise training may influence the observed effects on resting-state brain connectivity in different brain regions. For instance, Voss, Prakash et al. [54] suggested that exercise training increased connectivity in the default mode network (DMN) (included regions: the posterior cingulate, ventral and superior frontal medial cortices, and bilateral lateral occipital, middle frontal, hippocampal and parahippocampal, and middle temporal cortices) after 24 weeks (6 months), with additional changes emerging in the frontal-parietal network at 48 weeks (12 months) as compared to those in an active control group. Similarly, Prehn, Lesemann et al. [45] implemented exercise training of moderate length (24 weeks) in a sample of overweight older adults, and observed that a 24-week (6-month) moderate aerobic exercise program significantly increased connectivity between the dorsolateral prefrontal cortex (dlPFC) and superior parietal gyrus/precuneus compared to those in stretching and toning group. Similarly to a previous review [31], findings from this review reveal exercise training for 24 weeks increased connectivity in resting state networks.

When evaluating session time, six of the included studies found brain function changes following intervention in resting-state fMRI [45,54,55] and task-evoked fMRI [50,52,56] after a short session time (<45 min). For instance, Wu, Tang et al. [56] demonstrated that a Tai Chi training program lasting 30 min per session increased activation in the left SFG and the right MFG during a task-switching paradigm. Such beneficial effects were corroborated by Voelcker-Rehage, Godde et al. [53] following moderate session time (45–50 min) of aerobic and coordination exercise intervention. However, Flodin, Jonasson et al. [51] examined hippocampal functional connectivity after a 24-week aerobic training intervention of 30–60 min sessions, but did not find that the experimental group differed from the control group at post-test. Thus, we suggest that the length of the exercise training session should be consistent to understand the association between session time of exercise training and brain function.

Few studies in this review discussed volume and progression of exercise training, making it difficult to understand the effects of change in exercise training on brain function. Despite a wealth of evidence demonstrating that exercise interventions benefit functional brain outcomes, the dose–response relationship was not straightforward. Therefore, future studies are needed to provide a better understanding of the exercise training characteristics. Similarly, future research can quantify the total amount of exercise (e.g., combined exercise length, frequency, session time, or intensity) to better inform proper dosing guidelines of exercise training on brain structure and function.

### 4.4. Limitations

To the best of our knowledge, this is the first systematic review that has considered the characteristics of the exercise training program in an effort to understand the underlying factors of the exercise exposure on brain outcomes in older adults. Several limitations, however, should be considered. First, given the state of the field, only a limited number of studies were included in the systematic review. As such, the results should be interpreted with caution. Second, several of the included studies were flagged because they may have contained threats to internal validity. Third, we only collected studies that were published in the English language, thus potentially leading to issues related to publication bias.

### 4.5. Conclusions

The present review strengthens the field by systematically investigating the relationship between exercise training characteristics and brain outcomes. This review furthers our ability to provide a prescription for training that has the potential to improve brain structure and function in older adults. Importantly, we used the FITT-VP principal to guide our descriptive reporting of the benefits of exercise on brain outcomes. Findings from the present systematic review show that older adults involved in exercise training may derive benefits to brain health, as reflected in intervention-induced changes in brain structure and function, with such benefits dependent upon the dose of the exercise intervention. The heterogeneity across study methodology including allocation concealment and blinding of participants and assessors may limit the comparability of the findings across studies. Accordingly, we suggest that well-designed methodologies and reporting are necessary to further delineate the characteristics of the exercise intervention that lead to beneficial effects on brain structure and function in older adults.

## Figures and Tables

**Figure 1 jcm-09-00914-f001:**
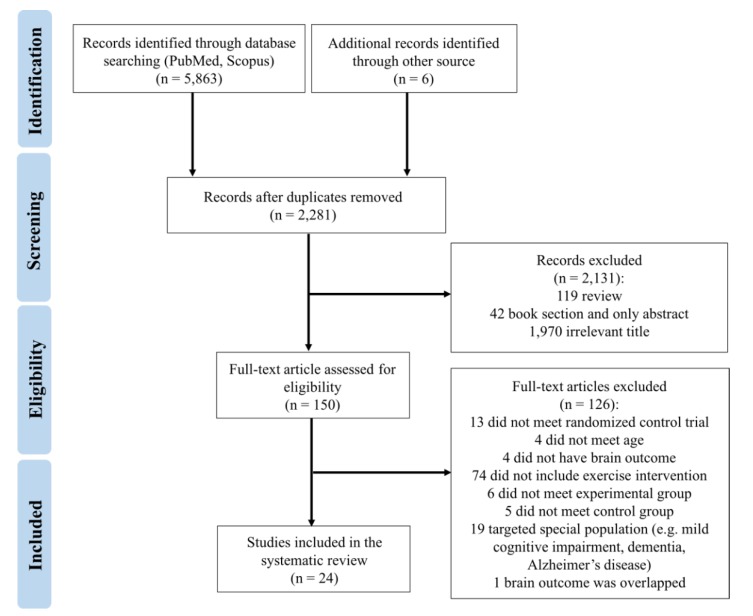
Flow diagram of studies included through the review process according to the Preferred Reporting Items for Systematic Reviews and Meta-Analyses (PRISMA).

**Figure 2 jcm-09-00914-f002:**
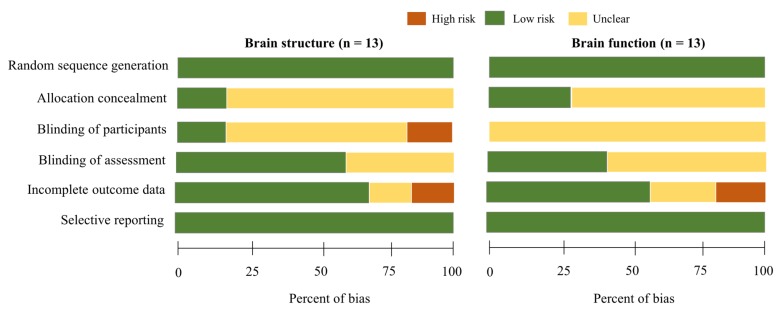
Risk of bias graph indicating the percentage of bias for all criteria in the included studies.

**Table 1 jcm-09-00914-t001:** Summary of experimental studies examining the effect of exercise intervention on brain structure.

Citation	Characteristics (*n*, % Female, Age)		Exercise Prescription	Main Finding	
	Experimental Group	Control Group		Region	Comparison
Best, Chiu et al. [43]	Resistance exercise once weekly (*n* = 52, 100%, 69.5 ± 2.7) Resistance exercise twice weekly (*n* = 54, 100%, 69.4 ± 3.0)	Balance-and-tone training (*n* = 49, 100%, 70.0 ± 3.3)	*F*: 1 or 2d/w *I*: 7RM (vigorous) *TM*: 52 weeks/60 min *TP*: Resistance exercise *V*: NR *P*: NR	Whole brain	*TD*: Resistance exercise once weekly group showed reduced cortical white matter volume atrophy at two-year follow-up but not at one-year follow-up.
Burzynska, Jiao et al. [42]	Dance group (*n* = 49, 76%, 65.88 ± 4.70) Walking group (*n* = 40, 68%, 64.98 ± 4.00)	Active control group (*n* = 43, 67%, 66.72 ± 4.65)	*F*: 3 d/w *I*: NR *TM*: 24 weeks/60 min *TP*: Dance and aerobic exercise *V*: NR *P*: NR	Twenty regions were selected	*BD*: Dance group increased RD and MD in fornix compared to walking and active control group; walking group decreased FA in fornix compared to active control group.
Colcombe, Erickson et al. [36]	Exercise group (*n* = NR, NR, 65.5)	Control group (*n* = NR, NR, 66.9)	*F*: 3d/w *I*: 60–70% HRR (vigorous) *TM*: 24 weeks/60 min *TP*: Aerobic exercise *V*: NR *P*: 40–50%~60–70% HRR	Whole brain	*TD*: Exercise group showed increased grey and white matter volume in prefrontal and temporal cortices.
Erickson, Voss et al. [47]	Aerobic exercise group (*n* = 60, NR, 67.6 ± 5.81)	Stretching control group (*n* = 60, NR, 65.5 ± 5.44)	*F*: NR *I*: 60-75% HRR (vigorous) *TM*: 48 weeks/40 min *TP*: Aerobic exercise *V*: NRP: 50–60%~60–75% HRR, 10~40 min *P*: 50–60%~60–75% HRR, 10~40 min	Hippocampus Caudate nucleus Thalamus	*TD*: Aerobic exercise group showed increased right and left hippocampal volume.
Jonasson, Nyberg et al. [40]	Aerobic group (*n* =29, 52%, 68.40 ± 2.54)	Control group (*n* = 29, 59%, 68.97 ± 2.91)	*F*: 3d/w *I*: 80% HR_max_ (vigorous) *TM*: 24 weeks/30–60 min *TP*: Aerobic training *V*: NR *P*: 40~80% HR_max_	Hippocampus dlPFC VPC ACC	*TD*: Neither group showed differences in cortical thickness.
Liu-Ambrose, Nagamatsu et al. [44]	Resistance exercise once weekly (*n* = 54, 100%, 69.5 ± 2.7) Resistance exercise twice weekly (*n* = 52, 100%, 69.4 ± 3.0)	Balance-and-tone training (*n* = 49, 100%, 70.0 ± 3.3)	*F*: 1 or 2d/w *I*: 7RM (vigorous) *TM*: 48 weeks/60 min *TP*: Resistance exercise *V*: NR *P*: NR	Whole brain	*GD*: Both exercise groups showed reductions in whole brain atrophy compared to balance-and-tone training group.
Maass, Düzel et al. [38]	Aerobic exercise group (*n* = 21, 52%, 68.8 ± 4.5)	Relax/stretching group (*n* = 19, 58%, 67.9 ± 4.1)	*F*: 3d/w *I*: 65% VO_2 VAT_ (vigorous) *TM*: 12 weeks/30 min *TP*: Aerobic exercise *V*: NR *P*: NR	Hippocampus	*TD*: Neither group showed changes in hippocampal volume.
Matura, Fleckenstein et al. [39]	Exercise group (*n* = 29, 42%, 73.3 ± 5.5)	Waiting control group (*n* = 24, 52%, 77.0 ± 8.1)	*F*: 3d/w *I*: 64 VO_2max_ (vigorous) *TM*: 12 weeks/30 min *TP*: Aerobic exercise *V*: NR *P*: NR	Whole brain	*TD*: Exercise group did not show changes in cortical grey matter volume.
Niemann, Godde et al. [41]	Cardiovascular training group (*n* = 17, 71%, 68.24 ± 2.61) Coordination training group (*n* = 19, 68%, 69.63 ± 5.10)	Control group (*n* = 13, 54%, 68.77 ± 2.56)	*F*: 3d/w *I*: NR *TM*: 48 weeks/45–60 min *TP*: Aerobic and coordination exercise *V*: NR *P*: NR	Hippocampus	*TD*: Both exercise groups showed increases in hippocampal volume.
Prehn, Lesemann et al. [45]	Aerobic exercise group (*n* = 11, 36%, 69)	Stretching and toning group (*n* = 18, 56%, 65)	*F*: 2d/w *I*: 80% AT (moderate) *TM*: 24 weeks/30 min *TP*: Aerobic exercise *V*: NR *P*: 20~30 min	Whole brain	*TD*: Neither group showed significant changes in grey matter volume.
Rosano, Guralnik et al. [48]	Physical activity group (*n* = 10, NR, 74.9 ± 4.4)	Health education group (*n* =16, NR, 76.8 ± 6.1)	*F*: NR *I*: Self-report (moderate) *TM*: 48 weeks/NR *TP*: Combined exercise *V*: NR *P*: NR	Hippocampus	*GD*: Physical activity group showed greater left and right hippocampal volume and left cornu ammonis compared with health education group.
Tao, Liu et al. [46]	Tai Chi Chuan group (*n* = 21, NR, 62.38 ± 2.07) Baduanjin group (*n* = 16, NR, 62.18 ± 2.02)	Control group (*n* = 24, NR, 60.16 ± 1.88)	*F*: 5d/w *I*: NR *TM*: 12 weeks/60 min *TP*: Tai Chi *V*: NR *P*: NR	Whole brain	*TD*: Tai Chi Chuan group increased grey matter volume in the left insula, left putamen, left parahippocampus/hippocampus, left amygdala, and left ITG as compared with control group; Baduanjin group increased grey matter volume in the insula, left hippocampus, left amygdala, bilateral putamen as compared to control group.
Voss, Heo et al. [37]	Walking group (*n* = 35, 69%, 65.17 ± 4.40)	Stretching group (*n* = 35, 60%, 64.57 ± 4.46)	*F*: 3d/w *I*: 60–75% HR_max_ (vigorous) *TM*: 48 weeks/40 min *TP*: Aerobic exercise *V*: NR *P*: 50–60~60–75% HR_max_, 10~40 min	Whole brain	*TD*: The walking group showed no significant effects on FA, AD, and RD.

Note: F = frequency; I = intensity; TM = length/session time; TP = type; V = volume; P = progression; NR = not reported; TD: time difference; GD = group difference; VAT = ventilatory anaerobic threshold; FA = fractional anisotropy; AD = axial diffusivity; RD = radial diffusivity; ACC = anterior cingulate cortex; dlPFC = dorsolateral prefrontal cortex; VPC = ventrolateral prefrontal cortex; ITG = inferior temporal gyrus.

**Table 2 jcm-09-00914-t002:** Summary of experimental studies examining the effect of exercise intervention on resting-state brain functioning.

Citation	Characteristics of Study (*n*, % Female, Age)		Exercise Prescription	Main Finding
	Experimental Group	Control Group		
Chapman, Aslan et al. [49]	Physical training group (*n* = 18, 72%, 64.0 ± 4.3)	Control group (*n* = 19, 74%, 64.0 ± 3.6)	*F*: 3d/w *I*: 50–75% HR_max_ (moderate) *TM*: 12 weeks/50 min *TP*: Aerobic exercise *V*: NR *P*: NR	*TD*: Neither group showed changes in CBF in the hippocampus. *BD*: Physical training group after intervention increased activation in CBF in bilateral ACC compared to the control group.
Flodin, Jonasson et al. [51]	Aerobic exercise group (*n* = 30, 53%, 68.41 ± 2.59)	Control group (*n* = 25, 56%, 69.16 ± 3.01)	*F*: 3d/w *I*: NR *TM*: 24 weeks/30–60 min *TP*: Aerobic exercise *V*: NR *P*: NR	*TD*: Neither group showed changes in functional connectivity after intervention.
Maass, Düzel et al. [38]	Aerobic exercise group (*n* = 21, 52%, 68.8 ± 4.5)	Relax/stretching group (*n* = 19, 58%, 67.9 ± 4.1)	*F*: 3d/w *I*: 65% VO_2 VAT_ (vigorous) *TM*: 12 weeks/30 min *TP*: Aerobic exercise *V*: NR *P*: NR	*TD*: Aerobic exercise group showed a significant change in hippocampal perfusion (i.e., CBF and CBV).
Prehn, Lesemann et al. [45]	Aerobic exercise group (*n* = 11, 36%, 69)	Stretching and toning group (*n* = 18, 56%, 65)	*F*: 2d/w *I*: 80% AT (moderate) *TM*: 24 weeks/30 min *TP*: Aerobic exercise *V*: NR*P*: 20~30 min	*GD*: Aerobic exercise group increased functional connectivity between dlPFC and superior parietal gyrus/precuneus as compared to stretching and toning group.
Tao, Chen et al. [57]	Tai Chi Chuan group (*n* = 21, 62%, 62.38 ± 4.55) Baduanjin group (*n* = 15, 60%, 62.33 ± 3.88)	Control group (*n* = 25, 76%, 59.76 ± 4.83)	*F*: 5d/w *I*: NR *TM*: 12 weeks/60 min *TP*: Tai Chi *V*: NR *P*: NR	*GD*: Tai Chi Chuan group significantly increased amplitude of low-frequency and low-4 band in right dlPFC as compared to control group; Baduanjin group significantly increased amplitude of low-frequency and low-4 band in the bilateral medial PFC as compared to control group.
Tao, Chen et al. [58]	Tai Chi Chuan group (*n* = 21, 62%, 62.38 ± 4.55) Baduanjin group (*n* = 15, 60%, 62.33 ± 3.88)	Control group (*n* = 25, 76%, 59.76 ± 4.83)	*F*: 5d/w *I*: NR *TM*: 12 weeks/60 min *TP*: Tai Chi *V*: NR *P*: NR	*GD*: Tai Chi Chuan group decreased functional connectivity in the left SFG, left dorsal anterior cingulate, and rostral ACC as compared to control group; Baduanjin group decreased resting-state functional connectivity in the left putamen/insula as compared to control group
Tao, Liu et al. [59]	Tai Chi Chuan group (*n* = 21, 62%, 62.38 ± 4.55) Baduanjin group (*n* = 15, 60%, 62.33 ± 3.88)	Control group (*n* = 25, 76%, 59.76 ± 4.83)	*F*: 5d/w *I*: NR *TM*: 12 weeks/60 min *TP*: Tai Chi *V*: NR *P*: NR	*GD*: Tai Chi Chuan group increased functional connectivity between bilateral hippocampus and right medial PFC and left medial PFC as compared to control group; Tai Chi Chuan group increased functional connectivity between left and right hippocampus as compared to control group
Voss, Erickson et al. [55]	Walking group (*n* = 30, 73%, 67.3 ± 5.8)	Flexibility, toning, balance group (*n* = 35, 71%, 65.4 ± 5.2)	*F*: 3d/w *I*: 60–75% HR_max_ (moderate) *TM*: 48 weeks/40 min *TP*: Aerobic exercise *V*: NR *P*: 50–60~60–75% HR_max_, 10~40 min	The study did not provide interventional results.
Voss, Prakash et al. [54]	Aerobic walking group (*n* = 30, 73%, 67.30 ± 5.80)	Control group (*n* = 35, 71%, 67.30 ± 5.24)	*F*: 3d/w *I*: 60–75% HRR (vigorous) *TM*: 48 weeks/40 min *TP*: Aerobic exercise *V*: NR *P*: 50–60%~60-75% HRR, 10~40 min	*TD*: Aerobic walking exercise group showed connection between right anterior PFC and PFC at 48 weeks into the intervention.

Note: F = frequency; I = intensity; TM = length/session time; TP = type; V = volume; P = progression; NR = not reported; TD: time difference; GD = group difference; AT = anaerobic threshold; CBF = cerebral blood flow; CBV = cerebral blood volume; SFG = superior frontal gyrus; ACC = anterior cingulate cortex; PFC = prefrontal cortex; IPG = inferior parietal gyrus; dlPFC = dorsolateral prefrontal cortex.

**Table 3 jcm-09-00914-t003:** Summary of experimental studies examining the effect of exercise intervention on task-evoked brain functioning.

Citation	Characteristics of Study (*n*, % Female, Age)		Exercise Prescription	Task	Main Finding
	Experimental Group	Control Group			
Colcombe, Kramer et al. [50]	All (*n* = 29, 62% female, 65.60 ± 5.66) Aerobic group (NR)	Control group (NR)	*F*: 3d/w *I*: 60–70% HRR (vigorous) *TM*: 24 weeks/40–45 min *TP*: Aerobic exercise *V*: NR *P*: 40–50%~60–70% HRR, 10–15~40–45 min	Ericksen flanker task	*GD*: Aerobic exercise group showed a significantly higher activation in MFG, SFG, SPL and reduced activation in ACC compared to the control group.
Nocera, Crosson et al. [52]	Spin group (*n* = 16, 63%, 69.7 ± 6.34)	Balance group (*n* = 14, 43%, 72.09 ± 6.43)	*F*: 3d/w *I*: 50~75% HRR (moderate-to-vigorous) *TM*: 12 weeks/45 min *TP*: Aerobic exercise *V*: NR *P*: 20~45 min	Semantic verbal fluency	*GD*: Spin group decreased activation in ITG and angular gyrus as compared to the balance group.
Voelcker-Rehage, Godde et al. [53]	Cardiovascular training group (*n* = 17, 71%, 68.47 ± 3.06) Coordination training group (*n* = 16, 63%, 71.13 ± 4.59)	Control group (*n* = 11, 55%, 69.27 ± 3.29)	*F*: 3d/w *I*: 60% VO_2peak_ (moderate) *TM*: 48 weeks/45–50 min *TP*: Aerobic and coordination exercise *V*: NR *P*: 35~45–50 min	Ericksen flanker task	*TD*: Cardiovascular training decreased activation in the MFG, the left ACC, the left parahippocampal gyrus, and the right STG, and the MTG; coordination training increased activation in the IFG, thalamus, caudate, and the SPL. *GD*: The two exercise training groups showed no changes, but the control group increased activation in the right MFG, the left anterior and right posterior cingulate, the right parahippocampal gyrus, the right STG, and the right lentiform nucleus.
Wu, Tang et al. [56]	Tai Chi Chuan group (*n* = 16, 81%, 64.9 ± 2.8)	Control group (*n* = 15, 100%, 64.9 ± 3.2)	*F*: 3d/w *I*: 65.4% HR_max_ (vigorous) *TM*: 12 weeks/30 min *TP*: Tai Chi *V*: NR *P*: NR	Task-switching	*TD*: Tai Chi Chuan group showed increased activation in the left SFG and the right MFG in switch condition as compared to non-switch; control group decreased activation in the left SFG, the right MFG, and left IPG.

Note: F = frequency; I = intensity; TM = length/session time; TP = type; V = volume; P = progression; NR = not reported; TD: time difference; GD = group difference; MFG = middle frontal gyrus; SFG = superior frontal gyrus; SPL = superior parietal lobules; ACC = anterior cingulate cortex; IFG = inferior frontal gyrus; STG = superior temporal gyrus; ITG = inferior temporal gyrus; MTG = middle temporal gyrus.

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
