# Peer review of "The Effect of Exercise Training on Brain Structure and Function in Older Adults: A Systematic Review Based on Evidence from Randomized Control Trials"

_jcm, 2020, doi:10.3390/jcm9040914_

Round 1

Reviewer 1 Report

Chen (2020): The Effect of Exercise Training on Brain Structure and Function in Older Adults: A Systematic Review Based on Evidence from Randomized Control Trials

 In the submitted manuscript, the authors have conducted a systematic review to compile findings from studies examining the effects of exercise on brain structure and function. Given the vast number of studies out there, and the existing contradictory findings, I think this is important work, with its novelty laying in its attempt to address questions regarding the frequency and intensity of interventions. I have a couple of major points and a few additional recommendations which I think might improve this manuscript.

Major:

 My main concern is that the manuscript presents the results from task-based fMRI and resting state studies together– can they really be considered together? Even just within task fMRI, is there a basis or precedence for pooling from findings across various tasks? What is being interpreted as “facilitated brain function in resting-state”? Could the authors please elaborate on their rationale for this.

  • I think it would be beneficial for the authors to be clearer on the outcome that is being extracted from each paper. If its group differences (intervention vs. control) or changes between pre- post. I think it would be clearer/more consistent if same type of comparison was focused on as outcome, or, alternatively, for the phrasing to be more specific when the outcomes are presented. For instance, Niemann et al (2014) found no significant difference between groups, but did find that hippocampal volume increased over time in both groups. If the coordination exercises are being considered as the control group in the review (which is what I’ve understood from the inclusion criteria), then it’s the group difference that is of interest, not the pre-and post.

Minor:

  • There have been a few systematic reviews and meta-analyses on the relationship between exercise and brain structures (e.g. white matter: Sexton et al 2016; hippocampus: Firth et al 2017), some which have found no evidence of such effects. I think the introduction would benefit from reflecting this.
  • Could the authors define their meaning of brain health? If it’s just measures of brain structure/connectivity, specify.
  • Could the authors please include more details on inclusion criteria for the fMRI outcomes – did the type of task matter?
  • What is meant by ‘general care’ in the inclusion criteria?
  • Could the authors elaborate on ‘volume’ and ‘progression’ from the extraction process? (lines 29/41)
  • Could the authors please elaborate on what was included as ‘exercise’? Were dance interventions excluded? If so, Burzynska’s study (2017) wasn’t picked up.
  • The bullet points and middle-formatting make the text in the cells harder to read.
  • Could the authors include n, female and mean age ± SD for all groups? Missing in some (e.g. Colcombe, n for Best). Or state “NR and unable to obtain from authors”. The table could be improved by presenting the demographics of each group in a separate column. I think splitting up region and findings into two separate columns would also make the table easier to navigate.
  • In Table 2, the task of the task-based fMRI should be specified.
  • In lines 300/301, the authors state: “These findings suggest that exercise frequency may not moderate exercise training effects on brain structure outcomes.”. It is unclear where this interpretation comes from, as it is following a summary of studies applying only moderate (3-4 times/week) intervention frequencies. This sentence probably needs to be moved to the end of that paragraph, so that it comes after the studies using other frequencies are mentioned.

Author Response

Reviewer # 1

In the submitted manuscript, the authors have conducted a systematic review to compile findings from studies examining the effects of exercise on brain structure and function. Given the vast number of studies out there, and the existing contradictory findings, I think this is important work, with its novelty laying in its attempt to address questions regarding the frequency and intensity of interventions. I have a couple of major points and a few additional recommendations which I think might improve this manuscript.

Dear Reviewer # 1

     Thank you very much for your positive review and enthusiasm for our work. Your helpful recommendations and encouragement have significantly improved the quality of our manuscript.

     To more easily identify your suggested changes, we have provided point-by-point responses to the issues you have raised and have highlighted modifications in green. We appreciate the time that you have devoted to reviewing our manuscript and sincerely hope that our modifications will meet your approval.

Major

1. My main concern is that the manuscript presents the results from task-based fMRI and resting state studies together– can they really be considered together? Even just within task fMRI, is there a basis or precedence for pooling from findings across various tasks? What is being interpreted as “facilitated brain function in resting-state”? Could the authors please elaborate on their rationale for this.

Response: Thank you for the suggestion. We agree that two approaches should be addressed separately and have edited the text and the tables accordingly (please refer to Table 1, 2, and Discussion).

2. I think it would be beneficial for the authors to be clearer on the outcome that is being extracted from each paper. If its group differences (intervention vs. control) or changes between pre- post. I think it would be clearer/more consistent if same type of comparison was focused on as outcome, or, alternatively, for the phrasing to be more specific when the outcomes are presented. For instance, Niemann et al (2014) found no significant difference between groups, but did find that hippocampal volume increased over time in both groups. If the coordination exercises are being considered as the control group in the review (which is what I’ve understood from the inclusion criteria), then it’s the group difference that is of interest, not the pre-and post.

Response: Thank you for pointing this out. We agree and have grouped the outcomes from group and time in the Table (please refer to Table 1, 2, 3) and further elucidated the group- and time-difference outcomes throughout the manuscript. In the case of Niemann et al (2014), coordination exercise was not recognized as a control group or comparison group in the manuscript and we extracted the effects of two exercise trainings (aerobic exercise, coordination exercise) on brain outcomes to compare to the control group as an interventional outcome.

Minor:

1. There have been a few systematic reviews and meta-analyses on the relationship between exercise and brain structures (e.g. white matter: Sexton et al 2016; hippocampus: Firth et al 2017), some which have found no evidence of such effects. I think the introduction would benefit from reflecting this.

Response: Thank you for the suggestion. The two reviews were included in the Introduction (please see page 02, line 66-69).

2. Could the authors define their meaning of brain health? If it’s just measures of brain structure/connectivity, specify.

Response: Thank you for the suggestion. As you suggested, we have revised the text and specifically defined brain health as brain structure and function throughout the manuscript (please refer to page 1, line 36-37, page 02, line 60, and page 17, line 415).

3. Could the authors please include more details on inclusion criteria for the fMRI outcomes – did the type of task matter?

Response: Thank you for the suggestion. As you suggested, we have added more details on inclusion criteria for the fMRI outcomes (please refer to page 03, line 121-123).  

4. What is meant by ‘general care’ in the inclusion criteria?

Response: Thank you for the suggestion. For clearly addressing the criteria, we revised the text to health education and promotion (please refer to page 02, line 98).

5. Could the authors elaborate on ‘volume’ and ‘progression’ from the extraction process? (lines 29/41)

Response: Thank you for the suggestion. We have added information about the extracting process in volume and progression of exercise training (please refer to page 03, line 132-134).

6. Could the authors please elaborate on what was included as ‘exercise’? Were dance interventions excluded? If so, Burzynska’s study (2017) wasn’t picked up.

Response: Thank you for pointing it out. We have described the definition of exercise accordingly (please refer to page 02, line 94-95) and also added Burzynska et al (2017) to the manuscript (please refer to Table 1 and Discussion).

7. The bullet points and middle-formatting make the text in the cells harder to read.

Response: Thank you for the suggestion. We have revised the text accordingly (please refer to Table 1, 2, 3).

8. Could the authors include n, female and mean age ± SD for all groups? Missing in some (e.g. Colcombe, n for Best). Or state “NR and unable to obtain from authors”. The table could be improved by presenting the demographics of each group in a separate column. I think splitting up region and findings into two separate columns would also make the table easier to navigate.

Response: Thank you for pointing it out. We have revised the text accordingly (please refer to Table 1, 2, 3).

9. In Table 2, the task of the task-based fMRI should be specified.

Response: Thank you for the suggestion. We have specified the task (Please refer to Table 3)

10. In lines 300/301, the authors state: “These findings suggest that exercise frequency may not moderate exercise training effects on brain structure outcomes.”. It is unclear where this interpretation comes from, as it is following a summary of studies applying only moderate (3-4 times/week) intervention frequencies. This sentence probably needs to be moved to the end of that paragraph, so that it comes after the studies using other frequencies are mentioned.

Response: Thank you for the suggestion. We have moved the sentence to the end of the paragraph (please refer to page 14, line 299-300).

Reviewer 2 Report

This is a very nice review focusing on the FITT-VP principle to highlight patterns in existing RCTs of exercise in older adults. 

  • Why was an 8 week cutoff used for RCTs? It seems studies employing less than an 8-week exercise training program could still be considered as 'training' studies rather than acute exercise.
  • On page 3, it would be helpful to provide the definitions of volume and progression as I think of volume as an aspect of progression, but these terms seem to be considered as distinct constructs elsewhere in paper (e.g., 3.3.6 on pg.6, 3.4.6 on pg. 9)
  • Figure 2 is a nice way to present the bias assessment results. In addition to the overall results, it would be nice to also present this same figure broken down by structural vs. functional studies in order to highlight any discrepancies in study quality between the two main outcomes of interest in this review.
  • The discussion is a bit skimpy. For example some mention of the possible mechanisms underlying these structural and functional brain changes observed across studies wold be helpful, even if citing existing reviews.

Minor comments:

There are some subject-verb agreement issues and missing words in various sentences throughout (e.g., Abstract line 8), which were a little distracting. An additional, careful read through would be useful. 

Author Response

Reviewer # 2

This is a very nice review focusing on the FITT-VP principle to highlight patterns in existing RCTs of exercise in older adults. 

Dear Reviewer # 2

     Thank you very much for your positive review and enthusiasm for our work. Your helpful recommendations and encouragement have significantly improved the quality of our manuscript.

     To more easily identify your suggested changes, we have provided point-by-point responses to the issues you have raised and have highlighted modifications in yellow. We appreciate the time that you have devoted to reviewing our manuscript and sincerely hope that our modifications will meet your approval.

1. Why was an 8-week cutoff used for RCTs? It seems studies employing less than an 8-week exercise training program could still be considered as 'training' studies rather than acute exercise.

Response: Thank you for the comments. We set the search criterion following a recent systematic review and several meta-analyses that analyzed exercise training and cognition, which also excluded exercise training RCTs for less than 2 months (i.e., 8-weeks; Falck et al., 2019). Although none of the searched articles (n=149) conducted an exercise training RCT for less than or equal to 8 weeks, we removed the criterion in the review (please refer to page 02, line 93-94).

Reference: Falck, R.S., et al., Impact of exercise training on physical and cognitive function among older adults: a systematic review and meta-analysis. Neurobiol. Aging, 2019. 79: p. 119-130.

2. On page 3, it would be helpful to provide the definitions of volume and progression as I think of volume as an aspect of progression, but these terms seem to be considered as distinct constructs elsewhere in paper (e.g., 3.3.6 on pg.6, 3.4.6 on pg. 9).

Response: Thank you for the suggestion. The definitions of volume and progression have been added (please see page 02, line 65-66). In addition, as you suggested, we have revised the contents and further described the volume and progression of exercise training, respectively (please see page 06, line 201-205 and page 08, line 244-248).

3. Figure 2 is a nice way to present the bias assessment results. In addition to the overall results, it would be nice to also present this same figure broken down by structural vs. functional studies in order to highlight any discrepancies in study quality between the two main outcomes of interest in this review.

Response: Thank you for the suggestion. This is a good idea for clearly distinguishing the risk of bias from studies about brain structure and function. Therefore, we have revised this section and updated the figure (please see page 04, line 156-167 and page 05, Figure 2).

4. The discussion is a bit skimpy. For example, some mention of the possible mechanisms underlying these structural and functional brain changes observed across studies would be helpful, even if citing existing reviews.

Response: Based on this suggestion, we have added review articles to elucidate the possible mechanisms for the effects on structural and functional brain changes from exercise training (please see page 11, line 306-307, page 11, line 316-317, page 12, line 378-379).

Minor comments:

1. There are some subject-verb agreement issues and missing words in various sentences throughout (e.g., Abstract line 8), which were a little distracting. An additional, careful read through would be useful.

Response: Thank you for the comments. Grammatical errors were corrected throughout the manuscript by a native English speaker.

Reviewer 3 Report

Thank you for the invitation to review this interesting manuscript. The authors have examined exercise training using an exposure-based approach (FITT-VP) to characterize the dose-response associations between exercise training and brain structural and functional outcomes. This is an important question that has not been adequately addressed in prior studies. The authors provide a systematic review of existing literature across heterogeneous training modalities of various types, durations, and intensities.

  • While the review is interesting, there is a missed opportunity to more systematically quantify the total exercise exposure as it relates to brain changes. Did the authors consider attempting to quantify exercise exposure in some formal fashion, e.g. total weeks X sessions X mins/session?
  • It would seem that sensitivity analyses contrasting or within exercise modalities might also be useful. Some of the modalities used differ in their putative mechanistic pathways (e.g. tai-chi vs. running), so sensitivity analyses could provide useful information in this regard.

Author Response

Reviewer # 3

Thank you for the invitation to review this interesting manuscript. The authors have examined exercise training using an exposure-based approach (FITT-VP) to characterize the dose-response associations between exercise training and brain structural and functional outcomes. This is an important question that has not been adequately addressed in prior studies. The authors provide a systematic review of existing literature across heterogeneous training modalities of various types, durations, and intensities.

Dear Reviewer # 3

      Thank you very much for your positive review and enthusiasm for our work. Your helpful recommendations and encouragement have significantly improved the quality of our manuscript.

      To more easily identify your suggested changes, we have provided point-by-point responses to the issues you have raised and have highlighted modifications in blue. We appreciate the time that you have devoted to reviewing our manuscript and sincerely hope that our modifications will meet your approval.

1. While the review is interesting, there is a missed opportunity to more systematically quantify the total exercise exposure as it relates to brain changes. Did the authors consider attempting to quantify exercise exposure in some formal fashion, e.g. total weeks X sessions X mins/session? It would seem that sensitivity analyses contrasting or within exercise modalities might also be useful. Some of the modalities used differ in their putative mechanistic pathways (e.g. tai-chi vs. running), so sensitivity analyses could provide useful information in this regard.

Response: Thank you for the suggestion to provide a different approach to present the exercise volume. While you make a good point, aggregating across multiple exercise components (e.g., frequency, duration, intensity) does not fit the main purpose of this review, rather we examine the effects of exercise training on brain structure and function based on the FITT-VP principle that included studies specifically presented. To clarity the reader, we have provided the description (please refer to page 03, line 124-125). It is unfortunate that no study has examined specifically the issue of exercise volume, presenting the importance of further exploration. That is, we believe your recommendation is of value for future research and we added this suggestion to the limitations and future directions section to reflect your good point (please see page 13, line 391-394).

Round 2

Reviewer 1 Report

I thank the authors for their response, and am satisfied that all my comments have been addressed. 

Reviewer 3 Report

The authors have addressed my prior concerns and I have no additional comments at the present time.